# Changes in Treatment Patterns and Globe Salvage Rate of Advanced Retinoblastoma in Korea: Efficacy of Intra-Arterial Chemotherapy

**DOI:** 10.3390/jcm10225421

**Published:** 2021-11-20

**Authors:** Dong Hyun Lee, Jung Woo Han, Seung Min Hahn, Byung Moon Kim, Chuhl Joo Lyu, Sung Chul Lee, Dong Joon Kim, Christopher Seungkyu Lee

**Affiliations:** 1Department of Ophthalmology, Inha University School of Medicine, 27, Inhang-ro, Jung-gu, Incheon 22332, Korea; wizmeca@live.co.kr; 2Department of Ophthalmology, Yonsei University College of Medicine, 50-1, Yonseiro, Seodaemun-gu, Seoul 03722, Korea; 3Department of Pediatric Hematology and Oncology, Yonsei University College of Medicine, 50-1, Yonseiro, Seodaemun-gu, Seoul 03722, Korea; jwhan@yuhs.ac (J.W.H.); bluenile88@yuhs.ac (S.M.H.); cj@yuhs.ac (C.J.L.); 4Department of Radiology, Yonsei University College of Medicine, 50-1, Yonseiro, Seodaemun-gu, Seoul 03722, Korea; bmoon21@yuhs.ac; 5Department of Ophthalmology, Konyang University College of Medicine, 158, Gwanjeodongro, Seo-gu, Daejeon 35365, Korea; sunglee@yuhs.ac

**Keywords:** retinoblastoma, antineoplastic agents, intra-arterial chemotherapy, intravenous chemotherapy, intravitreal chemotherapy

## Abstract

(1) Background: To analyze changes in treatment patterns for advanced retinoblastoma over time and differences in globe salvage rates; (2) Methods: Retrospective, observational case-control study of 97 eyes of 91 patients with advanced retinoblastoma (Group D and E).; (3) Results: Patients were divided into two groups based on whether they were treated before or after intraarterial chemotherapy (IAC) was introduced in our center in 2010. Before 2010, primary treatment pattern was enucleation, which was performed in 57.6% of cases, whereas primary treatment pattern after 2010 was IAC combined with intravenous chemotherapy (IVC), which was performed in 78.1%. Intravitreal chemotherapy (IVitC) has been performed to treat vitreous and subretinal seeding since 2015. The 5-year globe salvage rate of IVC alone was 24.0% for Group D and 0% for Group E, whereas that of IVC–IAC was 50.4% for Group D and 49.7% for Group E. Whether IVitC was performed or not did not significantly contribute to globe salvage rate. There was one metastatic death in the IVC alone group.; (4) Conclusions: Primary treatment pattern changed from enucleation to IAC-based treatment, which can now save nearly half of eyes with advanced retinoblastoma with excellent safety profile and survival rate.

## 1. Introduction

Retinoblastoma is the most common primary intraocular malignant tumor in children. Patients who develop metastatic disease often exhibit histopathological high-risk features including tumor invasion into the optic nerve and/or uvea [1,2]. Advanced tumors are marked with these high-risk features; Group D and E retinoblastoma shows 15 to 17% high-risk features and 24% to 50% high risk features, respectively [3,4]. Thus, patients with advanced Group D or E retinoblastoma have been generally recommended to have enucleation for fear of tumor involvement of the central nervous system or hematogenous spread of the disease to other organs. Carboplatin-based systemic intravenous chemotherapy (IVC) is commonly used to induce the chemoreduction of retinoblastoma [5,6,7]. However, treatment results of IVC on advanced retinoblastoma are not satisfactory, as local recurrence rate is high, resulting in eventual enucleation [8,9]. Recently introduced intra-arterial chemotherapy (IAC), which is a treatment strategy to inject anticancer drugs at the ostium of the ophthalmic artery, has shown promising results for the conservation of the eyeball, even those with advanced retinoblastoma, but by IAC alone, there still exists the concern for the development of metastatic disease [10,11,12].

Since 2010, we have treated advanced retinoblastoma patients with IAC combined with IVC. In the present study, we analyzed how the globe salvage rate of advanced Group D and E retinoblastoma has changed after the introduction of IAC. In addition, treatment patterns and outcomes (e.g., recurrence and metastasis) before and after the introduction of IAC were also comprehensively analyzed.

## 2. Materials and Methods

This is a retrospective, interventional case series. The medical records of retinoblastoma patients treated at Severance Hospital, Yonsei University College of Medicine, from August 1985 to July 2020 were reviewed. This study was conducted in accordance with the tenets of the Declaration of Helsinki and was approved by the Institutional Review Board of the Severance Hospital (IRB approval number: 4-2020-0596). Informed consent was waived. Eyes with either group D or E retinoblastoma based on the International Classification of Retinoblastoma (ICRB) were included. All patients were followed up for at least 1 year from the time of diagnosis. The demographic information, clinical findings, and treatment methods were reviewed and analyzed. A comprehensive ocular examination was conducted under general anesthesia for diagnosis and regular check-up several times.

The IVC–IAC protocol is as follows. IVC consisting of vincristine (1.5 mg/m^2^, day 1) carboplatin (200 mg/m^2^, day 1–2), etoposide (150 mg/m^2^), and cyclosporine [13] (12 mg/kg, day 1–2) (CVE) was performed as a first-line therapy. The IAC technique with fluoroscopic guidance has been previously described [14]. The IAC medication was melphalan (2.5 mg for patients <6 months, 3 mg for 6months—1 year, 4 mg for 1–3 years, 5 mg for >3 years), topotecan (0.5 mg for patients <1year, 1 mg for >1year), and carboplatin (25 mg for patients <1 year, 30 mg for 1–3 years, 40 mg for >3 years). IAC was first performed with melphalan monotherapy from 2010 to 2014. Since 2015, IAC has primarily been performed with melphalan and topotecan in combination. Carpoplatin was used during the period when melphalan was unavailable in Korea. For primary IAC, IAC was performed 3 weeks after IVC, then alternate IVC–IAC was repeated [14]. A systemic IVC with CVE regimen was repeated for 6–8 cycles at 4-week intervals or at 5/6 week interval if IAC was performed in-between IVC cycles. If the response to primary treatment was unsatisfactory, second line IVC including vincristine (1.5 mg/m^2^, day 1), doxorubicin (45 mg/m^2^, day 1), cyclophosphamide (500 mg/m^2^, day 1–3) (VaDdrC), and/or additional IAC were performed. Prior to the introduction of CVE regimen in about 1995 in our center, enucleation or external beam radiation therapy (EBRT) were primarily performed without IVC for treatment of retinoblastoma patients.

Intravitreal chemotherapy (IVitC), as an adjuvant therapy for IAC, has been performed on some patients since 2015. The IVitC medication included melphalan, topotecan, or methotrexate. IVitC was primarily performed with melphalan or topotecan, and methotrexate was used in selective cases (2 cases) where clinical response to melphalan/topotecan IVitC was deemed suboptimal. Under general anesthesia, intravitreal injection was performed through pars plana at 2–3 mm from the limbus, depending on the age of the patients. Intravitreous melphalan (25–30 µg in 0.04–0.08 mL), topotecan (10–20 µg in 0.04 mL), or methotrexate (400 µg in 0.05 mL) was prepared in the operating room under sterile conditions. Through the pars plana route with a 30-G needle, intravitreous injection was performed.

Focal therapies, including transpupillary thermotherapy (TTT) with an infrared diode laser (810 nm), laser photocoagulation with argon green laser (532 nm), cryotherapy with triple-freeze-thaw technique, and brachytherapy with a Ruthenium-106 eye applicator (BEBIG Isotopen und Medizintechnik GmbH, Berlin, Germany) were performed under anesthesia as needed to consolidate tumor or treat recurrence.

Survival analysis was performed to analyze globe salvage rate, recurrence-free survival, and metastasis-free survival before and after 2010. In addition, the globe salvage rate of IVC–IAC treatment and IVC alone was also comparatively analyzed. SPSS statistics v25.0 (IBM Inc., Armonk, NY, USA) was used for statistical analysis. Descriptive analysis was performed, and Student’s *t*-test, chi-square test, or Fisher’s exact test were performed. The globe retention duration of the patients was analyzed, and the globe salvage rate was confirmed through Kaplan–Meier survival analysis. Log rank test was used to check whether the survival curve was different.

## 3. Results

### 3.1. Demographics and Treatment Information

Ninety-seven eyes of 91 patients were included in this study. Twenty-eight (30.8%) patients had retinoblastoma in both eyes, of which, six (6.6%) patients had a bilateral group D or E tumor. Forty-nine (53.8%) patients were female. Forty-seven eyes were the right eye and 50 eyes were the left eye. The mean age at diagnosis was 21.3 ± 14.8 months. Patients were followed up for an average of 96.9 ± 70.1 months. There were 41 (42.3%) group D eyes and 56 (57.7%) group E eyes. Patients received an average of 10.6 ± 5.2 cycles of IVC treatments, 4.1 ± 2.0 IAC treatments, and 8.8 ± 8.0 IVitC treatments. After the introduction of IAC, the number of IVC treatments was significantly higher after 2010 than before 2010 (11.2 ± 5.4 vs. 8.7 ± 3.9, *p* < 0.001). A total of 202 IAC treatments were performed. Twenty-three (23.7%) patients received IVitC treatments. A total of 193 IVitC were performed. TTT was performed in 38 (39.2%, 3.8 ± 3.7 times) patients. Laser photocoagulation was performed in 19 (19.6%, 2.3 ± 1.9 times) patients. External cryotherapy was performed in 17 (17.5%, 2.2 ± 1.0 times) patients. Extraction of the cataract that obscured the fundus visualization was performed in 5 (5.2%) patients, all of whom eventually underwent enucleation due to evident or suspicious tumor recurrence, and none developed extraocular extension or metastases. Three (3.1%) patients underwent vitrectomy as an alternative to enucleation because parents strongly refused enucleation. However, all these eyes were subsequently enucleated due to failed tumor control, and no patient developed metastatic disease. Nine (9.3%) patients underwent adjuvant external beam radiation therapy. One (1.0%) patient received brachytherapy to treat tumor recurrence following IVC–IAC, but eventually underwent enucleation. (Table 1).

Thirty-three eyes were diagnosed before 2010 and 64 eyes after 2010. There was no difference in demographical features, including age at diagnosis, sex, and ICRB group. The group diagnosed before 2010 had a longer follow-up period (*p* < 0.001). The mean number of IVCs was significantly higher in the group diagnosed after 2010 (*p* < 0.001). More TTT and fewer EBRT were performed in the group diagnosed after 2010 (*p* = 0.001 and 0.006, respectively). The proportion of patients who underwent enucleation was lower in the group diagnosed after 2010 (*p* < 0.001) with higher recurrence rate (*p* = 0.039) (Table 1).

### 3.2. Distribution of Patients and the Reason for Enucleation

Before 2010, primary enucleation was performed in 7 (46.7%) eyes in Group D and 12 (66.7%) eyes in Group E, and IVC was performed in the remaining eyes. All these eyes, except for one Group D eye, eventually underwent enucleation. After 2010, primary enucleation was performed in 2 (7.7%) eyes in Group D and 7 (18.4%) eyes in Group E. IVC alone was performed in 1 (3.8%) eye of Group D and 4 (10.5%) eyes of Group E, of which all Group E eyes were eventually enucleated. IVC–IAC was performed in 23 (88.5%) eyes of Group D and 27 (71.1%) eyes of Group E. Secondary enucleation rate for Group D and Group E after 2010 were 41.7% and 45.2%, respectively. The reason for enucleation is described in Table 2.

### 3.3. Overall 5-Year Globe Salvage, Recurrence-Free, and Metastasis-Free Rates

The overall 5-year globe salvage rate was 28.9% (95% confidence interval: 18.1–38.9). The 5-year globe salvage rates before 2010 and after 2010 were 3.2% (95% confidence interval: 0.2–14.1) and 44.5% (95% confidence interval: 30.7–57.4), respectively (*p* < 0.001). The overall 5-year recurrence-free rate was 63.4% (95% confidence interval: 52.6–72.4). The 5-year recurrence-free rates before 2010 and after 2010 were 81.5% (95% confidence interval: 63.4–91.2) and 54.7% (95% confidence interval: 41.4–66.2), respectively (*p* = 0.006). The overall 5-year metastasis-free rate was 98.8% (95% confidence interval: 91.8–99.8). The 5-year metastasis-free rates before 2010 and after 2010 were 96.9% (95% confidence interval: 79.8–99.6 (before 2010) and 100.0% (95% confidence interval: 100–100), respectively (*p* = 0.619 Figure 1). The results of comparing survival analysis of IVC alone and IVC–IAC are as follows: The 5-year globe salvage rate of IVC alone was 24.0% for Group D (95% confidence interval: 3.4–51.3) and 0% for Group E (95% confidence interval: 0–0), while that of IVC–IAC was 50.4% for Group D (95% confidence interval: 27.9–73.9) and 49.7% for Group E (95% confidence interval: 29.3–67.2). The overall 5-year globe salvage rates of advanced retinoblastoma (Group D and E) for IVC and IVC–IAC were 17.5% (95% confidence interval: 3.3–41.0) and 49.4% (95% confidence interval: 33.1–63.7), respectively (*p* = 0.024, Figure 2). Whether IVitC was performed or not did not contribute significantly to the globe salve rate in the present study (*p* = 0.852).

### 3.4. Complications Associated with IVC and IAC Treatments

Complications associated with IVC and IAC treatments are described in Table 3. Neutropenic fever was the most common complication associated with systemic IVC. The most common ocular complication associated with IAC was eyelid swelling, which occurred in 11 (22.9%) eyes. There was one case of acute hemorrhagic retinopathy following intravitreal melphalan. One patient who was referred to us for management of recurred tumor after 26 cycles of IVC elsewhere underwent two sessions of IAC and subsequent enucleation in our institute, but eventually died from second primary acute myelomonocytic leukemia. There were no other deaths of patients.

## 4. Discussion

Complete tumor control with IVC alone is often limited in advanced retinoblastomas, and IAC has been recently introduced to improve management in them [15,16]. A recent meta-analysis on the treatment effect of IAC showed the globe salvage rate of 63.3% in Group A–C eyes with retinoblastoma and 35% in Group D–E eyes [17]. Before we began IAC in our practice, the primary treatment for advanced retinoblastoma (Group D–E eyes) was enucleation, which was performed in 57.6% of eyes upon presentation. Primary IVC was largely unsuccessful, as 92.9% of eyes eventually underwent enucleation. After IAC was introduced, the primary enucleation rate for advanced retinoblastoma decreased to 14.1% and primary treatment pattern became IVC–IAC, which was performed in 78.1%. We found that the 5-year globe salvage rate for IVC–IAC was 49.4% in Group D–E eyes in the present study.

IAC has shown rapid tumor control and a higher globe retention effect in advanced retinoblastoma compared to IVC treatment [18]. IAC can be safely and effectively applied as a primary therapy to infants less than 3 months old [19] and has not been associated with increased metastatic risk, raising a controversy over whether additional IVC is necessary in the ear of ICA [12,20]. Chen et al. reported that combining IVC and IAC did not show additional benefit to primary IAC treatment alone in local tumor control or 3-year survival of patients with advanced retinoblastoma treatment [21]. However, Xu et al. found that a combined IVC and IAC group showed better tumor control, higher eyeball salvage rate, and lower levels of serum markers (vascular endothelial growth factor, neuron specific enolase, livin, and survivin), compared to the IAC alone group [22]. Because advanced retinoblastoma often presents high-risk histopathologic features that may promote the risks of systemic metastasis, we have managed advanced retinoblastoma patients with IVC–IAC treatment, preferably in an alternate fashion. Combining IVC can have practical advantage in some centers where IAC requires longer preparatory time; IVC can be promptly initiated while preparing for IAC under general anesthesia with anesthesiologists and interventional radiologists. However, whether alternate IVC–IAC treatment or IAC as a second line treatment has benefits compared to IAC alone in advanced retinoblastoma patients warrants further study.

IVitC is effective for the treatment of advanced retinoblastoma with vitreous or subretinal seeding [23,24], and is widely used as an adjuvant therapy [25]. In this study, IVitC did not significantly affect globe salvage rate in the patients treated with IVC–IAC. However, it should be noted that those eyes that received IVitC also tend to have a greater number of IAC, compared to eyes that did not receive IVitC, suggesting that IVitC-treated eyes were more advanced eyes.

There are some limitations in this study. The small sample size and retrospective study design may affect the statistical significances of the present study. Treatment selection bias is possible, because attempts to salvage eyeballs would have been made more frequently after IAC became available. Furthermore, before the CVE multidrug chemotherapy regimen was employed at our center in about 1995, enucleation or EBRT were primarily performed. However, to the best of knowledge, the present study is the largest case series of advanced retinoblastoma treated with IAC in Korea, showing changes in treatment pattern toward IAC-based treatment and subsequent improvement in globe salvage success rates.

## 5. Conclusions

In conclusion, after IAC was introduced, treatment pattern changed from primary enucleation to IVC–IAC in our center, which has saved nearly half of the eyeballs of advanced retinoblastoma patients and has an excellent survival rate. IVitC effectively treated vitreous and subretinal seeding, which has likely contributed to enhanced globe salvage rate with IAC.

## Figures and Tables

**Figure 1 jcm-10-05421-f001:**
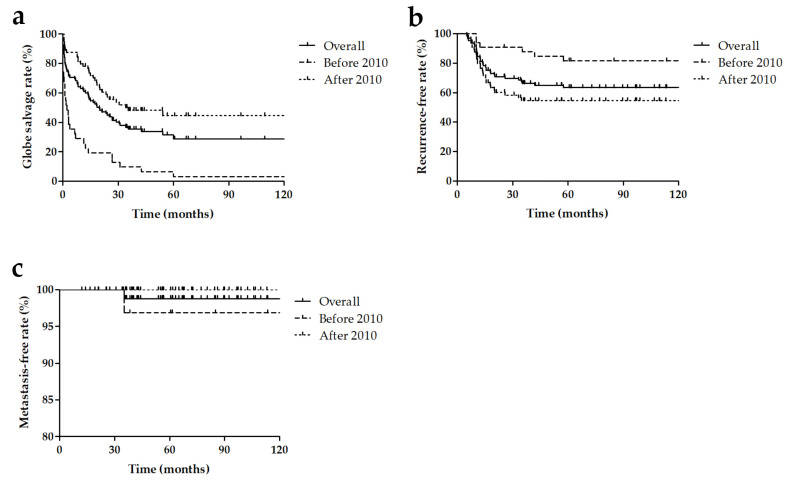
Kaplan–Meier curves showing the globe-salvage rates (**a**), recurrence-free rates (**b**), and metastasis-free rates (**c**) of advanced retinoblastoma based on whether patients were diagnosed before or after 2010.

**Figure 2 jcm-10-05421-f002:**
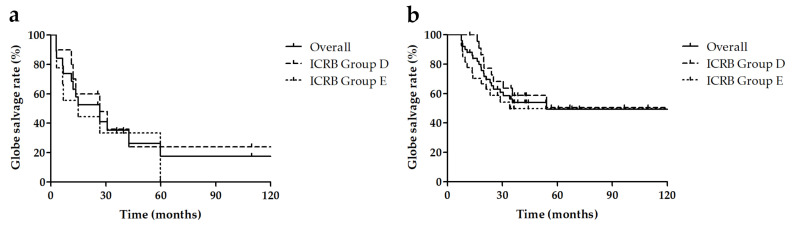
Kaplan–Meier curves showing globe salvage rates of advanced retinoblastoma treated with intravenous chemotherapy (IVC) alone (**a**) and with combined IVC and intraarterial chemotherapy (IAC) (**b**). ICRB: International Classification of Retinoblastoma; IVC: intravenous chemotherapy; IAC: intra-arterial chemotherapy.

**Table 1 jcm-10-05421-t001:** Demographics, clinical features, and treatment outcome of 97 eyes of 91 retinoblastoma patients according to time periods in which the diagnosis was made.

	1985–2009 (Eyes = 33)	2010–2020 (Eyes = 64)	Total	*p*-Value ^1^
Bilateral tumors (Any grade) ^2^	9 (29.0%)	19 (31.7%)	28 (30.8%)	0.817
Bilateral tumors (Group D/E) ^2^	2 (6.5%)	4 (6.7%)	6 (6.6%)	0.999
ICRB group				0.670
Group D	15 (45.5%)	26 (40.6%)	41 (42.3%)	
Group E	18 (54.5%)	38 (59.4%)	56 (57.7%)	
Total number of IAC	-	202	202	-
Melphalan	-	116 (57.4%)	116 (57.4%)	-
Melphalan + Topotecan	-	80 (39.6%)	80 (39.6%)	-
Topotecan	-	1 (0.5%)	1 (0.5%)	-
Topotecan + Carboplatin	-	5 (2.5%)	5 (2.5%)	-
Total number of IVitC	-	193	193	-
Melphalan	-	109 (56.5%)	109 (56.5%)	-
Topotecan	-	74 (38.3%)	74 (38.3%)	-
Methotrexate	-	10 (5.2%)	10 (5.2%)	-
Additional treatment				
TTT	5 (15.2%)	33 (51.6%)	38 (39.2%)	0.001
Laser photocoagulation	3 (9.1%)	16 (25.0%)	19 (19.6%)	0.103
External cryotherapy	3 (9.1%)	14 (21.9%)	17 (17.5%)	0.161
Cataract surgery	1 (3.0%)	4 (6.3%)	5 (5.2%)	0.659
Vitrectomy	0 (0.0%)	3 (4.7%)	3 (3.1%)	0.549
EBRT	7 (21.2%)	2 (3.1%)	9 (9.3%)	0.006
Brachytherapy	0 (0.0%)	1 (1.6%)	1 (1.0%)	0.999
Treatment outcomes				
Enucleation	32 (97.0%)	33 (51.6%)	65 (67.0%)	<0.001
Recurrence	6 (18.2%)	28 (43.8%)	34 (35.1%)	0.039
Metastasis	1 (3.0%)	0 (0.0%)	1 (1.0%)	0.340

^1^ Comparison between patients diagnosed before 2010 and after 2010. ^2^ Number of patients, not eyes. Student’s *t*-test, chi-square test, or Fisher’s exact test was used as appropriate. EBRT: external beam radiation therapy; ICRB: International Classification of Retinoblastoma; IAC: intraarterial chemotherapy; IVC: intravenous chemotherapy; IVitC: intravitreal chemotherapy, SD: standard deviation; TTT: transpupillary therapy.

**Table 2 jcm-10-05421-t002:** Summary of the distribution of patients and the reason for enucleation in advanced retinoblastoma patients.

	1985–2009		2010–2020	
	Group D N = 15	Group E N = 18	Group D N = 26	Group E N = 38
Primary enucleation	7 (46.7%)	12 (66.7%)	2 (7.7%)	7 (18.4%)
IVC	8 (53.3%)	6 (33.3%)	1 (3.8%)	4 (10.5%)
IVC + IAC	0 (0%)	0 (0%)	23 (88.5%)	27 (71.1%)
Secondary enucleation	7 (87.5%)	6 (100.0%)	10 (41.7%)	14 (45.2%)
Clinically evident tumor	6 (85.7%)	2 (33.3%)	5 (50.0%)	9 (64.3%)
Total RD with suspicious recurrence	1 (14.3%)	2 (33.3%)	1 (10.0%)	2 (14.3%)
Painful eye with phthisis bulbi	0 (0.0%)	2 (33.3%)	4 (40.0%)	3 (21.4%)

RD: retinal detachment; IAC: intraarterial chemotherapy; IVC: intravenous chemotherapy.

**Table 3 jcm-10-05421-t003:** Adverse events related to treatments in retinoblastoma patients who received intravenous chemotherapy and intraarterial chemotherapy.

Adverse Events	Number (%)
Systemic chemotherapy-related	
Fever	4 (8.3%)
Hypotension	1 (2.1%)
Seizure	1 (2.1%)
Intraarterial chemotherapy-related	
Eyelid swelling	11 (22.9%)
Erythema	7 (14.6%)
Conjunctival injection	6 (12.5%)
Ptosis	2 (4.2%)
Strabismus	2 (4.2%)

## Data Availability

The datasets generated during and/or analyzed during the current study are available from the corresponding author on reasonable request.

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
