# Peer review of "Changes in Treatment Patterns and Globe Salvage Rate of Advanced Retinoblastoma in Korea: Efficacy of Intra-Arterial Chemotherapy"

_jcm, 2021, doi:10.3390/jcm10225421_

Round 1
Reviewer 1 Report
The aim of this work was to analyze the effect of different treatments on the ocular globe salvage. The methodology consisted in a retrospective study of 91 patients with stage D and E retinoblastoma (ICRB). The patients were classified into two groups according to the treatment received, with and without intra-arterial therapy. The results of the treatments were analyzed with respect to the proportion of saved eyes and recurrence-free and metastasis-free survival. The authors conclude that intra-arterial chemotherapy made it possible to save eyes in patients with advanced retinoblastoma.
The design of the treatments is well detailed and the results were statistically analyzed. The subject of this manuscript: retinoblastoma treatment, is frequently published and the methodology used is not completely original, except in the design of the protocol. The presentation of the manuscript is accurate and the analysis of the results is well explained for their understanding and reproducibility. The conclusions are supported by the obtained results
Author Response
Reviewer 1
The aim of this work was to analyze the effect of different treatments on the ocular globe salvage. The methodology consisted in a retrospective study of 91 patients with stage D and E retinoblastoma (ICRB). The patients were classified into two groups according to the treatment received, with and without intra-arterial therapy. The results of the treatments were analyzed with respect to the proportion of saved eyes and recurrence-free and metastasis-free survival. The authors conclude that intra-arterial chemotherapy made it possible to save eyes in patients with advanced retinoblastoma.
The design of the treatments is well detailed and the results were statistically analyzed. The subject of this manuscript: retinoblastoma treatment, is frequently published and the methodology used is not completely original, except in the design of the protocol. The presentation of the manuscript is accurate and the analysis of the results is well explained for their understanding and reproducibility. The conclusions are supported by the obtained results
- Thank you for your kind review.
Reviewer 2 Report
Although the study reported in this article appears to show substantial benefits of combined intravenous chemotherapy (IVC) and intraarterial chemotherapy (IAC) for cases of ICRB groups D and E compared with IVC without intraarterial chemotherapy, there are issues of lack of correspondence of stated objectives with presented results, inclusion of some inappropriate terminology, and apparent combination of cases treated by planned combined IVC and IAC with cases treated by planned IVC and subsequent IAC after failure of local tumor control was observed. More specific comments are contained in the accompanying document.
Major Criticisms
Failure to mention IVC regimen employed prior to 1995
To this reviewer’s knowledge, the carboplatin-vincristine-etoposide (CVE) multidrug chemotherapy regimen did not start to become used widely as treatment for retinoblastoma until about 1995. In view of this, the authors need to state what regimen or regimens of chemotherapy were employed at their center in children with retinoblastoma they encountered and treated with initial chemotherapy between 1980 and 1995.
Inconsistent stated Objectives versus reported Results of the reported study
The last paragraph of the article’s Introduction states that the authors’ principal objective was to summarize and report results of a retrospective “review [of] our experience with advanced Group D and E retinoblastoma managed with IVC and IAC in an effort to salvage eye globe.” This statement suggests that the only patients whose results would be reported in this article would be ones who underwent both IVC and IAC. The last sentence mentions a secondary objective that was to report “how globe salvage rate changed as treatment modality included IAC”. The latter of these two objectives seems inconsistent with the first stated objective if only cases treated by both IVC and IAC for group D and E retinoblastoma were to be reported. The authors need either to (1) restate their objectives more comprehensively or (2) limit the results they report only to cases managed by combined IVC and IAC.
In the Materials and Methods section of this article, the authors do not mention any intention to compare subgroups of cases with Group D or E retinoblastoma treated prior to 2010 with cases treated after the start of 2010. Yet, the Results section that follows presents just such a comparison. The authors need to mention this intention in their Materials and Methods section and provide a clear rationale for subdividing their cases in this manner.
In the Results section of this article, the results reported by the authors include those of cases managed by primary enucleation and primary IVC alone as well as those of cases managed by both IVC and IAC. Inclusion of such cases seems inconsistent with the stated principal objective of the study mentioned above. In this section, the authors also appear to have combined cases treated initially by planned IVC and IAC with cases treated initially by IVC that were also treated subsequently by IAC once unsatisfactory local tumor control was identified. If the authors’ real intention was to report the results of planned combined IVC and IAC for eyes with Group D and E retinoblastoma, then inclusion of cases treated by IAC as a supplement to initial IVC after local treatment failure was detected seems inappropriate.
Minor Criticisms
Inappropriate references to ICRB classification as staging
The authors refer to the assignment of cases of retinoblastoma to ICRB groups D and E as “staging” (see first paragraph of Material and Methods section and column 1 in Table 1). Staging refers to a comprehensive whole body evaluation of a patient with cancer that predicts the likelihood of the patient’s death from that malignant neoplasm. The ICRB classification system is not a staging system but rather a prognostic system predictive of likelihood of globe salvage following initial treatment intended to eradicate intraocular retinoblastoma and be eye preserving. The authors need to eliminate any reference to ICRB classification as staging.
Inappropriate or missing column headers in Tables 1 and 2
The header for column 1 in table 1 is
Laterality
(Right/Left)
Yet, the principal items presented in this column are the names of “variables” the authors have summarized.
The header for columns 2 and 3 in table 1 is
No./Mean + SD
Yet, the actual data presented under these column headers include counts, counts and corresponding percentages, count of involved right eyes divided by count of involved left eyes, mean values and standard deviations of those mean values, and minimum & maximum values. In this reviewer’s opinion, authors should never present data of more than one kind under any single column header in any table. The authors need to rework Tables 1 and 3 extensively to address this issue.
No descriptive column header is provided for column 4 in Table 1.
Inappropriate centration of inclusive year column headers
In Table 2, the principal column header specifying inclusive years (i.e., 1985-2010 and 2010-2020) should be centered between columns 2 and 3 and between columns 3 and 4 and not centered over columns 2 and 4.
Inappropriate row header to report maximal and minimal values of certain variables
The authors present “(range)” as one of the row headers in Table 1. The “range” of a numerical variable is defined mathematically as the difference between the maximal and minimal values of that variable. As such, range is a single value. What the authors are reporting as “(range)” is actually a listing of the minimal and maximal values of the specified variable.
Overlapping years of subgroups
The reported intervals in this article (i.e., 1980-2010 and 2010-2020) include an overlapping year (2010). The authors need to report either 1980-2009 versus 2010-2020 or 1980-2010 versus 2011-2020.
Author Response
Reviewer 2
Although the study reported in this article appears to show substantial benefits of combined intravenous chemotherapy (IVC) and intraarterial chemotherapy (IAC) for cases of ICRB groups D and E compared with IVC without intraarterial chemotherapy, there are issues of lack of correspondence of stated objectives with presented results, inclusion of some inappropriate terminology, and apparent combination of cases treated by planned combined IVC and IAC with cases treated by planned IVC and subsequent IAC after failure of local tumor control was observed. More specific comments are contained in the accompanying document.
Major Criticisms
Point 1. Failure to mention IVC regimen employed prior to 1995
To this reviewer’s knowledge, the carboplatin-vincristine-etoposide (CVE) multidrug chemotherapy regimen did not start to become used widely as treatment for retinoblastoma until about 1995. In view of this, the authors need to state what regimen or regimens of chemotherapy were employed at their center in children with retinoblastoma they encountered and treated with initial chemotherapy between 1980 and 1995.
- Response 1. Thank you for your valuable comments. In our center, we started chemotherapy for retinoblastoma patients since about 1997 with CVE. Before then, enucleation and external beam radiotherapy were main treatments. We added this information in Materials and Methods and included this point as a limitation in discussion.
(page 2, 4th paragraph, line 88-90) “Prior to the introduction of CVE regimen in about 1995 in our center, enucleation or external beam radiation therapy (EBRT) were primarily performed without IVC for treatment of retinoblastoma patients.”
(page 8, 2nd paragraph, line 266-267) “Furthermore, before CVE multidrug chemotherapy regimen was employed at our center in about 1995, enucleation or EBRT were primarily performed.”
Point 2. Inconsistent stated Objectives versus reported Results of the reported study
The last paragraph of the article’s Introduction states that the authors’ principal objective was to summarize and report results of a retrospective “review [of] our experience with advanced Group D and E retinoblastoma managed with IVC and IAC in an effort to salvage eye globe.” This statement suggests that the only patients whose results would be reported in this article would be ones who underwent both IVC and IAC. The last sentence mentions a secondary objective that was to report “how globe salvage rate changed as treatment modality included IAC”. The latter of these two objectives seems inconsistent with the first stated objective if only cases treated by both IVC and IAC for group D and E retinoblastoma were to be reported. The authors need either to (1) restate their objectives more comprehensively or (2) limit the results they report only to cases managed by combined IVC and IAC.
- Response 2. Thank you for your sharp comment. As the reviewer recommended, we restated our objectives more comprehensively as follows in the introduction:
(page 2, 2nd paragraph, line56-59) “In the present study, we analyzed how the globe salvage rate of advanced Group D and E retinoblastoma has changed after the introduction of IAC. In addition, treatment patterns and outcomes (e.g. recurrence and metastasis) before and after the introduction of IAC were also comprehensively analyzed. we review our experience with advanced Group D and E retinoblastoma managed with IVC and IAC in an effort to salvage eye globe. Moreover, we analyzed how globe salvage rate changed as treatment modality included IAC.”
Point 3, In the Materials and Methods section of this article, the authors do not mention any intention to compare subgroups of cases with Group D or E retinoblastoma treated prior to 2010 with cases treated after the start of 2010. Yet, the Results section that follows presents just such a comparison. The authors need to mention this intention in their Materials and Methods section and provide a clear rationale for subdividing their cases in this manner.
- Response 3. Thank you for your comment. We added the intention for comparison in Materials and Methods as follows:
(page 3, 3rd paragraph, line 106-108) “Survival analysis was performed to analyze globe salvage rate, recurrence-free sur-vival, and metastasis-free survival before and after 2010. In addition, the globe salvage rate of IVC-IAC treatment and IVC alone was also comparatively analyzed.”
Point 4. In the Results section of this article, the results reported by the authors include those of cases managed by primary enucleation and primary IVC alone as well as those of cases managed by both IVC and IAC. Inclusion of such cases seems inconsistent with the stated principal objective of the study mentioned above. In this section, the authors also appear to have combined cases treated initially by planned IVC and IAC with cases treated initially by IVC that were also treated subsequently by IAC once unsatisfactory local tumor control was identified. If the authors’ real intention was to report the results of planned combined IVC and IAC for eyes with Group D and E retinoblastoma, then inclusion of cases treated by IAC as a supplement to initial IVC after local treatment failure was detected seems inappropriate.
- Response 4. Thank you for your another sharp comment. Because this is a retrospective review of all Group D and E retinoblastoma eyes to note changes in treatment patterns especially in terms of newly introduced IAC, heterogenous treatment pattern was a limitation. IAC was performed with intention to use it in either primary or secondary depending on cases, as the reviewer indicated. The primary intention of the present study was not to look at the result of primary IAC-IVC patients, but to see how treatment patterns changed toward using more IAC in treatment modality and how that changed the treatment results in terms of globe salvage rate. In this revision, we restated our study intension as such to be more comprehensive and clear as the reviewer indicated in another comment.
Minor Criticisms
Point 5. Inappropriate references to ICRB classification as staging
The authors refer to the assignment of cases of retinoblastoma to ICRB groups D and E as “staging” (see first paragraph of Material and Methods section and column 1 in Table 1). Staging refers to a comprehensive whole body evaluation of a patient with cancer that predicts the likelihood of the patient’s death from that malignant neoplasm. The ICRB classification system is not a staging system but rather a prognostic system predictive of likelihood of globe salvage following initial treatment intended to eradicate intraocular retinoblastoma and be eye preserving. The authors need to eliminate any reference to ICRB classification as staging.
- Response 5. Thank you for your comment. The words that mean staging have been removed and replaced as follows:
(page 2, 3rd paragraph, line 67-68) “Eyes with either group D or E retinoblastoma based on the International Classification of Retinoblastoma (ICRB) staging were included.”
(page 4, 1st paragraph, line 151-152) “There was no difference in demographical features including age at diagnosis, sex, and ICRB groupstaging.”
Point 6. Inappropriate or missing column headers in Tables 1 and 2
The header for column 1 in table 1 is
Laterality
(Right/Left)
Yet, the principal items presented in this column are the names of “variables” the authors have summarized.
The header for columns 2 and 3 in table 1 is
No./Mean + SD
Yet, the actual data presented under these column headers include counts, counts and corresponding percentages, count of involved right eyes divided by count of involved left eyes, mean values and standard deviations of those mean values, and minimum & maximum values. In this reviewer’s opinion, authors should never present data of more than one kind under any single column header in any table. The authors need to rework Tables 1 and 3 extensively to address this issue.
No descriptive column header is provided for column 4 in Table 1.
- Response 6. Thank you for your comment. Because there is too much information in the column of Table 1, which is inconvenient to read, only counts and corresponding percentages are left in the column, and the rest is described in the Result section as follows:
(page 3, 4th paragraph, line 120-121) “Forty-seven eyes were the right eye and 50 eyes were the left eye.”
(page 3, 4th paragraph, line 122) “Patients were followed up for an average of 96.9 ± 70.1 months.”
(page 3, 4th paragraph, line 124-126) “After the introduction of IAC, the number of IVC treatments was significantly higher after 2010 than before 2010 (11.2 ± 5.4 vs. 8.7 ± 3.9, p<0.001)”
(page 3, Table 1)
|
1985-2009 (eyes=33) |
2010-2020 (eyes=64) |
Total |
P-value1 |
No./ Mean ± SD |
No./ Mean ± SD |
|||
Laterality (Right/Left) |
16/17 |
31/33 |
47/50 |
0.999 |
Bilateral tumors (Any grade)2 |
9 (29.0%) |
19 (31.7%) |
28 (30.8%) |
0.817 |
Bilateral tumors (Group D/E)2 |
2 (6.5%) |
4 (6.7%) |
6 (6.6%) |
0.999 |
Male/Female2 |
13/18 |
29/31 |
42/49 |
0.830 |
Mean age at diagnosis, mo. (minimal – maximal values) |
24.1 ± 17.5 (1-74) |
19.9 ± 13.1 (2-61) |
21.3 ± 14.8 (1-74) |
0.190 |
Mean follow-up periods, mo. (minimal – maximal values) |
163.2 ± 77.0 (25-396) |
62.7 ± 31.3 (12-131) |
96.9 ± 70.1 (12-396) |
<0.001 |
ICRB groupstaging |
|
|
|
0.670 |
Group D |
15 (45.5%) |
26 (40.6%) |
41 (42.3%) |
|
Group E |
18 (54.5%) |
38 (59.4%) |
56 (57.7%) |
|
Treatments |
|
|
|
|
Mean IVC number (minimal – maximal values) |
8.7 ± 3.9 (2-16) |
11.2 ± 5.4 (2-26) |
10.6 ± 5.2 (2-26) |
<0.001 |
Mean IAC number (minimal – maximal values) |
- |
4.1 ± 2.0 (1-9) |
4.1 ± 2.0 (1-9) |
- |
Mean IVitC number (minimal – maximal values) |
- |
8.8 ± 8.0 (1-26) |
8.8 ± 8.0 (1-26) |
- |
Total number of IAC |
- |
202 |
202 |
- |
Melphalan |
- |
116 (57.4%) |
116 (57.4%) |
- |
Melphalan + Topotecan |
- |
80 (39.6%) |
80 (39.6%) |
- |
Topotecan |
- |
1 (0.5%) |
1 (0.5%) |
- |
Topotecan + Carboplatin |
- |
5 (2.5%) |
5 (2.5%) |
- |
Total number of IVitC |
- |
193 |
193 |
- |
Melphalan |
- |
109 (56.5%) |
109 (56.5%) |
- |
Topotecan |
- |
74 (38.3%) |
74 (38.3%) |
- |
Methotrexate |
- |
10 (5.2%) |
10 (5.2%) |
- |
Additional treatment |
|
|
|
|
TTT (Mean no. per patients) |
5 (15.2%, 7.8 ± 5.1) |
33 (51.6%, 3.2 ± 3.1) |
38 (39.2%, 3.8 ± 3.7) |
0.001 |
Laser photocoagulation (Mean no. per patients) |
3 (9.1%, 3.3 ± 2.1) |
16 (25.0%, 2.1 ± 1.8) |
19 (19.6%, 2.3 ± 1.9) |
0.103 |
External cryotherapy (Mean no. per patients) |
3 (9.1%, 2.7 ± 1.5) |
14 (21.9%, 2.1 ± 0.9) |
17 (17.5%, 2.2 ± 1.0) |
0.161 |
Cataract surgery |
1 (3.0%) |
4 (6.3%) |
5 (5.2%) |
0.659 |
Vitrectomy |
0 (0.0%) |
3 (4.7%) |
3 (3.1%) |
0.549 |
EBRT |
7 (21.2%) |
2 (3.1%) |
9 (9.3%) |
0.006 |
Brachytherapy |
0 (0.0%) |
1 (1.6%) |
1 (1.0%) |
0.999 |
Treatment outcomes |
|
|
|
|
Enucleation |
32 (97.0%) |
33 (51.6%) |
65 (67.0%) |
<0.001 |
Recurrence |
6 (18.2%) |
28 (43.8%) |
34 (35.1%) |
0.039 |
Metastasis |
1 (3.0%) |
0 (0.0%) |
1 (1.0%) |
0.340 |
Point 7. Inappropriate centration of inclusive year column headers
In Table 2, the principal column header specifying inclusive years (i.e., 1985-2010 and 2010-2020) should be centered between columns 2 and 3 and between columns 3 and 4 and not centered over columns 2 and 4.
- Response 7. Thank you for your comment. It was confirmed that the widths of 1985-2009 and 2010-2020 cells in the first row are slightly different. As suggested, we adjusted the cell spacing to put inclusive years in the center.
(page 5, Table 2)
|
1985-2009 |
2010-2020 |
||
Group D N=15 |
Group E N=18 |
Group D N=26 |
Group E N=38 |
|
Primary enucleation |
7 (46.7%) |
12 (66.7%) |
2 (7.7%) |
7 (18.4%) |
IVC |
8 (53.3%) |
6 (33.3%) |
1 (3.8%) |
4 (10.5%) |
IVC + IAC |
0 (0%) |
0 (0%) |
23 (88.5%) |
27 (71.1%) |
Secondary enucleation |
7 (87.5%) |
6 (100.0%) |
10 (41.7%) |
14 (45.2%) |
Clinically evident tumor |
6 (85.7%) |
2 (33.3%) |
5 (50.0%) |
9 (64.3%) |
Total RD with suspicious recurrence |
1 (14.3%) |
2 (33.3%) |
1 (10.0%) |
2 (14.3%) |
Painful eye with phthisis bulbi |
0 (0.0%) |
2 (33.3%) |
4 (40.0%) |
3 (21.4%) |
Point 8. Inappropriate row header to report maximal and minimal values of certain variables
The authors present “(range)” as one of the row headers in Table 1. The “range” of a numerical variable is defined mathematically as the difference between the maximal and minimal values of that variable. As such, range is a single value. What the authors are reporting as “(range)” is actually a listing of the minimal and maximal values of the specified variable.
- Response 8. Thank you for your comment. As you mentioned, it is correct to write “minimal and maximal values” instead of “range”. However, as in the recommendations above, Table 1 was simplified and all of ”range” were removed from this table as follows:
(page 3, Table 1)
|
1985-2009 (eyes=33) |
2010-2020 (eyes=64) |
Total |
P-value1 |
Bilateral tumors (Any grade)2 |
9 (29.0%) |
19 (31.7%) |
28 (30.8%) |
0.817 |
Bilateral tumors (Group D/E)2 |
2 (6.5%) |
4 (6.7%) |
6 (6.6%) |
0.999 |
ICRB group |
|
|
|
0.670 |
Group D |
15 (45.5%) |
26 (40.6%) |
41 (42.3%) |
|
Group E |
18 (54.5%) |
38 (59.4%) |
56 (57.7%) |
|
Total number of IAC |
- |
202 |
202 |
- |
Melphalan |
- |
116 (57.4%) |
116 (57.4%) |
- |
Melphalan + Topotecan |
- |
80 (39.6%) |
80 (39.6%) |
- |
Topotecan |
- |
1 (0.5%) |
1 (0.5%) |
- |
Topotecan + Carboplatin |
- |
5 (2.5%) |
5 (2.5%) |
- |
Total number of IVitC |
- |
193 |
193 |
- |
Melphalan |
- |
109 (56.5%) |
109 (56.5%) |
- |
Topotecan |
- |
74 (38.3%) |
74 (38.3%) |
- |
Methotrexate |
- |
10 (5.2%) |
10 (5.2%) |
- |
Additional treatment |
|
|
|
|
TTT |
5 (15.2%) |
33 (51.6%) |
38 (39.2%) |
0.001 |
Laser photocoagulation |
3 (9.1%) |
16 (25.0%) |
19 (19.6%) |
0.103 |
External cryotherapy |
3 (9.1%) |
14 (21.9%) |
17 (17.5%) |
0.161 |
Cataract surgery |
1 (3.0%) |
4 (6.3%) |
5 (5.2%) |
0.659 |
Vitrectomy |
0 (0.0%) |
3 (4.7%) |
3 (3.1%) |
0.549 |
EBRT |
7 (21.2%) |
2 (3.1%) |
9 (9.3%) |
0.006 |
Brachytherapy |
0 (0.0%) |
1 (1.6%) |
1 (1.0%) |
0.999 |
Treatment outcomes |
|
|
|
|
Enucleation |
32 (97.0%) |
33 (51.6%) |
65 (67.0%) |
<0.001 |
Recurrence |
6 (18.2%) |
28 (43.8%) |
34 (35.1%) |
0.039 |
Metastasis |
1 (3.0%) |
0 (0.0%) |
1 (1.0%) |
0.340 |
Point 9. Overlapping years of subgroups
The reported intervals in this article (i.e., 1980-2010 and 2010-2020) include an overlapping year (2010). The authors need to report either 1980-2009 versus 2010-2020 or 1980-2010 versus 2011-2020.
- Response 9. We apologize for the confusion and thank you for your comment. We changed the years as 1985-2009 versus 2010-2020 in the revision.
(page 3, Table 1)
|
1985-200910 (eyes=33) |
2010-2020 (eyes=64) |
Total |
P-value1 |
No./ Mean ± SD |
No./ Mean ± SD |
(page 5, Table 2)
|
1985-200910 |
2010-2020 |
||
Group D N=15 |
Group E N=18 |
Group D N=26 |
Group E N=38 |
|
Primary enucleation |
7 (46.7%) |
12 (66.7%) |
2 (7.7%) |
7 (18.4%) |
IVC |
8 (53.3%) |
6 (33.3%) |
1 (3.8%) |
4 (10.5%) |
IVC + IAC |
0 (0%) |
0 (0%) |
23 (88.5%) |
27 (71.1%) |
Secondary enucleation |
7 (87.5%) |
6 (100.0%) |
10 (41.7%) |
14 (45.2%) |
Clinically evident tumor |
6 (85.7%) |
2 (33.3%) |
5 (50.0%) |
9 (64.3%) |
Total RD with suspicious recurrence |
1 (14.3%) |
2 (33.3%) |
1 (10.0%) |
2 (14.3%) |
Painful eye with phthisis bulbi |
0 (0.0%) |
2 (33.3%) |
4 (40.0%) |
3 (21.4%) |
